# Ecosystem Service Evaluation and Multi-Objective Management of *Pinus massoniana* Lamb. Plantations in Guangxi, China

**Rongjian Mo [1,2], Yongqi Wang [1,2], Shulong Dong [1,2], Jiangming Ma [1,2,\*] and Yanhua Mo [2,\*]**

1. Key Laboratory of Ecology of Rare and Endangered Species and Environmental Protection, Guangxi Normal University, Ministry of Education, Guilin 541006, China
2. Guangxi Key Laboratory of Landscape Resources Conservation and Sustainable Utilization in Lijiang River Basin, Guangxi Normal University, Guilin 541006, China
* Correspondence: mjming03@gxnu.edu.cn (J.M.); moyanhua2019@mailbox.gxnu.edu.cn (Y.M.)

**Abstract:** Research on forest ecosystem service (ES) assessments is currently a topic of much interest in forest ecology combined with geography. Forests are the mainstay of terrestrial ecosystems and provide a wide range of welfare supports for humans. *Pinus massoniana* (*PM*) Lamb. is a major silvicultural timber species in southern China and plays an important role in meeting domestic timber demand as well as providing ESs. The assessment of the ESs of *PM* plantation forests is beneficial for their sustainable management. In this study, we used the woody biomass method, the InVEST water content model and the USLE, a generalized soil loss equation model to assess the values of four Ess, including wood supply, carbon sequestration and oxygen release, water conservation, and soil conservation, in *PM* plantations in the Guangxi Paiyangshan Forest Farm, which is a national *PM* seed base. A spectral clustering method was used to identify the ecosystem service clusters (i.e., partitions) in the case study area, and certain strategies were provided for different partitions to form a management strategy for the sustainable management of *PM* plantation forest ecosystems. This study showed that the value of each ES per hectare in the northern tropical pine plantation, ranked from the largest to the smallest, was water conservation; carbon sequestration and oxygen release; wood supply; and soil conservation, with the values of the wood supply in addition to carbon sequestration and oxygen release per hectare increasing with the age of the plantation. Based on the different service characteristics within the service clusters, the northern tropical *PM* plantation area was divided into wood supply, ecological nourishment and ecological restoration areas, which can focus more on wood supply and ecological nourishment.

**Keywords:** *Pinus massoniana* plantations; forest ecosystem services; ecosystem service cluster; multi-objective management strategy

## 1. Introduction

*PM* plantation forest resources are rich in timber production, fuel supply, turpentine provision, social security, ecological regulation and other service functions, but traditional forest management uses *PM* plantation forests in a single way, often focusing only on the utilization of timber production capacity and neglecting its supply capacities, such as those of ecological benefits and cultural services [1]. Therefore, it is particularly important to improve the awareness of managers and the public concerning the ESs of *PM* plantation forests in addition to maximizing the exploitation of the various service resources of *PM* plantation forests for the benefit of human beings.

Using the amount of ESs values as a measure, it is crucial to study the ecological and economic values of *PM* plantation forests in order to maintain their ecological security, improve regional environmental quality, provide economic value to timber, and carry out *PM* plantation forests. Zhou, P. et al. [2] proposed a rapid assessment method for ESs

based on productivity and biodiversity to provide a theoretical basis for the study of ES valuation. Kornatowska, B. et al. [3] reviewed the methods for estimating the value of forest ESs from the direction of economic production towards sustainable human well-being. Negi, G. C. S. [4] analyzed the complexity and challenges of quantifying and valuing ESs from academic and policy development perspectives. Paudyal, K. et al. [5] analyzed the values of ESs from 2005 to 2015 for four ES changes due to the increase in the number of acacia plantations in Central Vietnam, including carbon sequestration, sediment retention, water quantity and habitat. Wang, N. et al. [6] outlined the connotations of ES functions and the classification of their categories in addition to summarizing the progress of domestic and international research and the assessment methods for valuing ES functions. In recent years, there have been more studies on large-scale and multitype ESs, but there are fewer studies on the valuation of forest ESs and multi-objective management strategies. There is a lack of research on an ES value quantity assessment and multi-objective management strategy for *PM* plantation forests. This study mainly explored the ecosystem management strategy of *PM* plantation forests derived from the ES value quantity of the forest area in the northern tropical case study area, and it explored the spatial and temporal characteristics of four ES value quantities of wood supply, carbon sequestration and oxygen release, water conservation, and soil conservation in the case study area. The relationship characteristics among ESs were studied, the spectral clustering method was used to partition the ESs, respective multi-objective management strategies were summarized according to the characteristics of the different partitions, and quality as well as efficiency improvement strategies for *PM* plantation forests were proposed by combining the research results and the achievements of other scholars. This is beneficial to the sustainable development of *PM* plantation forests and provides some reference value for the assessment of the ES value amount and the formulation of service zoning and ecosystem management strategies for *PM* plantation forests in Guangxi.

## 2. Materials

### 2.1. Study Area

The Guangxi Paiyangshan Forest Farm (Figure 1), located in Ningming County, southwest Guangxi, China, near the border with China and Vietnam, is the only large state-owned forestry field directly under the Forestry Department of Guangxi Autonomous Region bordering Vietnam, with a total land area of 28,096.29 hm$^2$ and a forest land area of 27,462.28 hm$^2$. The geographical coordinates are 106°30′–107°15′ E, 21°15′–22°30′ N. The forestry field is a low mountainous landscape with an altitude of 200–800 m and a northern tropical monsoon climate. In this study, the Paiyangshan Forest Farm *PM* plantation is in the northern tropical case study area, which has an average annual temperature of 21.8 °C, an average annual rainfall of 1250–1700 mm, a rainy season from May to August, and a relative humidity of 82.5%. The annual number of sunshine hours is 1650.3 h, and the annual evaporation is 1423.3 mm. The soil is red soil, yellow-red soil, etc., with red soil being the main one. After years of afforestation, the native vegetation has been replaced by natural broad-leaved secondary forests and planted forests. The plantation forest tree species are mainly *PM*, *Eucalyptus robusta*, and *Illicium verum*.

### 2.2. Data Source

The basic data involved in this study include remote sensing data, forest resources type II survey data, socioeconomic data, meteorological data, and DEM data on the northern tropical case study area. After acquiring the above data, in order to facilitate statistical and spatial analyse all of the raster data were processed by rasterization and spatial resampling methods, and all raster data were unified with geographic coordinates, such as the WGS-84 coordinate system; the projection corresponds to Mercator coordinates with a spatial resolution of 30 × 30 m.

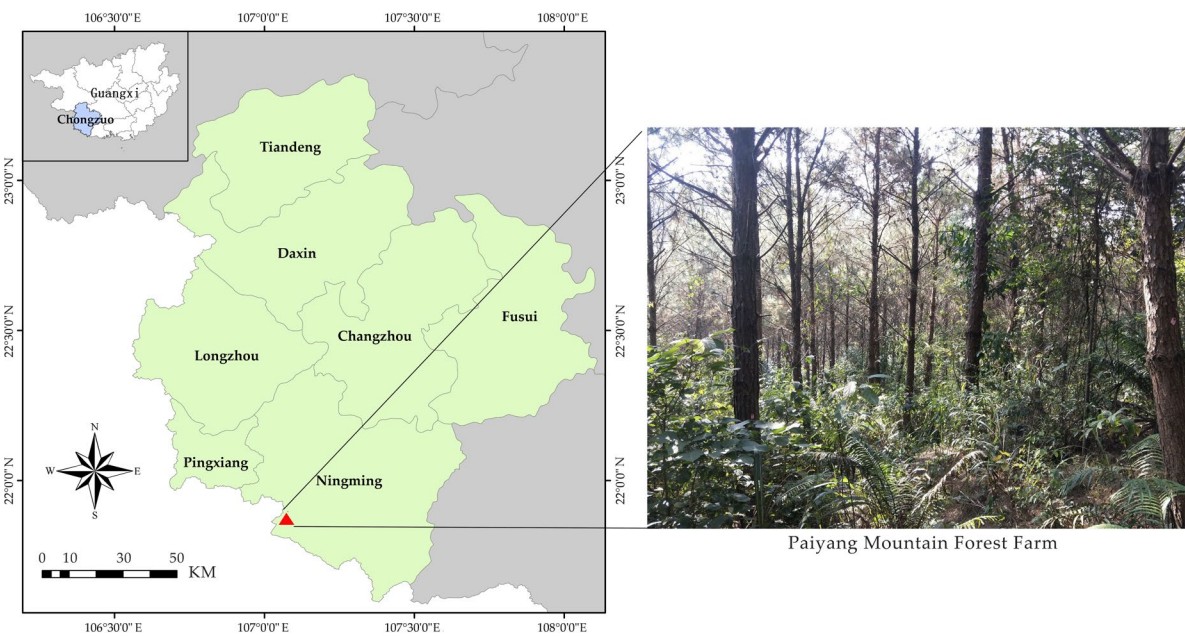

**Figure 1.** Distribution map of the northern tropical case study area.

### 2.2.1. Forest Resources Type II Survey Data

The forest resources type II survey data originated from forest resources inventory work, and the work is conducted every five years; thus, the forest resources type II survey data from 2009, 2013, and 2018 in the northern tropical case study area were obtained for this study. Considering that this study only explored the ecosystem services of *PM* plantation forests, the vector data and attribute data of the *PM* plantation forests were extracted for the study; these included the small group codes, small group areas, unit accumulation, slope, forest age, soil type, average diameter at breast height, and average tree height of the *PM* in small groups in the forest resources type II survey data. Based on the forest resources type II survey data, and according to the national dominant tree species age group classification table [7], *PM* plantation forests were classified into young (less than 10 years), mid-aged (11–20 years), mature (21–50 years) and over-mature (51 years or more) forests.

### 2.2.2. DEM Data

DEM (digital elevation model) data were downloaded from the geospatial data cloud (http://www.gscloud.cn/sources/ accessed on 15 April 2021), with a spatial resolution of 30 × 30 m. The downloaded DEM data were mosaicked, cropped, and puddle-filled using ArcGIS, where the cropping operation was performed by using the spatial extent of the *PM* plantation forests in each year in the case study area.

### 2.2.3. Vegetation Cover Data

Due to the cloudy and rainy weather in Guangxi, there was a great shadow in the image captures of the remote sensing satellites, such as optical sensors, and the remote sensing images of the study area needed to be screened in order to select the remote sensing images with small cloud coverage over the study area for the vegetation cover's interpretation. Landsat 4–5 and Landsat 8 satellite images in the time periods of 2009, 2013, and 2018 were screened (Table 1), and a total of three scenes with less than 10% cloud cover were screened. The remote sensing images were downloaded from the geospatial data cloud (http://www.gscloud.cn/sources accessed on 16 November 2022), and their spatial resolutions were all 30 × 30 with a WGS-84 projection.

**Table 1.** Landsat remote sensing image information.

| Article Code | Line Number | Data Identification | Imaging Date | Sensor Type |
|---|---|---|---|---|
| 126 | 045 | LT51260452009014BKT00 | 14 January 2009 | TM |
| 126 | 045 | LC81260452013361LGN00 | 27 December 2013 | OLI |
| 126 | 045 | LC08126045201810060111SC20200527100242 | 6 October 2018 | OLI |

The TM sensor is the sensor carried by the Landsat 4–5 satellite; it has seven bands and a coverage period of 16 days. The OLI sensor is the sensor carried by Landsat 8; it includes nine bands with a spatial resolution of 30 m, including a 15 m panchromatic band with an imaging width of 185 × 185 km.

### 2.2.4. Meteorological Data

The study area is mainly located in the southwestern part of Guangxi, and 40 meteorological station data points for all of the counties in Guangxi as well as counties in the surrounding provinces of Guangxi were downloaded and purchased from the China Meteorological Science Data Sharing Service Network. The data set included temperature data, rainfall data, and solar radiation data. In order to ensure that the data format for the model that was run was consistent, kriging interpolation was applied to the meteorological station data to form faceted raster data, and the geographic coordinates were unified as UTM WGS-84.

### 2.2.5. Soil Properties and Related Data Generation

The soil property data were mainly obtained from the Chinese soil data set of the World Soil Database (https://webarchive.iiasa.ac.at/Research/LUC/External-World-soil-database/HTML/ accessed on 15 April 2021) of the National Tibetan Plateau Scientific Data Center, with a spatial resolution of 1 km. The data set contained the soil gravel content (T_SAND), powder content (T_SILT), clay content (T_CLAY), soil bulk weight (T_REF_BULK), soil organic carbon content (T_OC), etc. The soil spatial data set was obtained by cropping the vector spatial data of the PM plantations in the case study area in each year.

### 2.2.6. Nutrient Contents of the Soil in PM Plantation Forests

The data on the soil nutrient content of *PM* plantations were obtained from the soil measurement indices completed by the group's project and from studies by other scholars [8–12] on soil nutrients in *PM* plantations (Table 2).

**Table 2.** Soil nutrients in *PM* plantations.

| Age Group | Soil Capacity (t/m³) | Soil Organic Matter (kg/t) | Soil Total Nitrogen (kg/t) | Soil Total Phosphorus (kg/t) | Soil Total Potassium (kg/t) |
|---|---|---|---|---|---|
| Young forest | 1.38 | 35.2386 | 1.70 | 4.53 | 0.097 |
| Mid-aged forest | 1.35 | 40.7381 | 2.07 | 0.54 | 0.143 |
| Mature forest | 1.33 | 46.4963 | 1.77 | 0.52 | 0.063 |
| Over-mature forest | 1.15 | 56.6162 | 1.37 | 0.63 | 0.110 |

## 3. Evaluation Methodology

This study mainly assessed the value quantity based on the calculation results of the wood supply, carbon sequestration and oxygen release, water conservation, and soil conservation material quality according to the Specification for the Assessment of Forest ES Functions (GB/T 38582-2020) [13]. In order to make the assessment results more realistically reflect the ESs of the northern tropical *PM* plantation forests, this study used data from the forest resources type II survey data on the northern tropical case study area as the basis, and it applied corrected remote sensing data combined with meteorological data and soil data to conduct the value quantity assessment of the four ESs. Among them, the carbon

sequestration price is usually adopted internationally at the Swedish carbon tax rate of 150 USD·t$^{-1}$ [14] (converted to 961.275 RMB·t$^{-1}$ at an exchange rate of 6.4085 USD to RMB; RMB is the legal tender of the People's Republic of China, and the unit is the RMB), and the other value quantity data were mainly derived from the prices recommended for use in the Specification for the Assessment of Forest ES Functions (GB/T 38582-2020).

### 3.1. Wood Supply Assessment

The data on the unit area storage volume and diameter at breast height of the small classes of *PM* plantation forests in the forest resources type II survey data were used, combined with the table on the economic timber yield of *PM* in Guangxi (Guangxi Forestry Branch and Guangxi Forestry Survey Institute, 1986). The uniform price of the different timber volumes of pine timber in Guangxi was used to make a value volume assessment of the wood supply capacity of *PM* plantation forests in the case study area, which is an estimation of the potential wood supply capacity of *PM* plantation forests.

Firstly, the amount of timber produced by each diameter order and the amount of short timber in a small group of a *PM* plantation were determined, after which the volume of the timber of the same diameter order was calculated to obtain the volume of timber for each diameter order in a small group. Finally, the volume of all timber species was calculated to obtain the total volume of the timber produced in a small group [15].

The market price of the pine timber in Guangxi was used to calculate the total forestry value of the *PM* plantations. The unit price of timber with a small head diameter above 26 cm was 900 RMB·m$^{-3}$; the unit price of timber between 18 and 24 cm was 750 RMB·m$^{-3}$; the unit price of timber between 4 and 16 cm was 520 RMB·m$^{-3}$; and the unit price of short small timber was 430 RMB·m$^{-3}$. The output value of the timber of this diameter was first calculated based on the output volume of the timber of each diameter order and the corresponding price, and the cumulative output value of each diameter order was calculated to obtain the total value of the timber provided by the small classes [15].

### 3.2. Carbon Sequestration and Oxygen Release Assessment

Based on the unit area accumulation data from forest resources type II survey data, the net productivity of the small groups of *PM* plantations was calculated by applying the woody source biomass method [16,17] to quickly and effectively derive a more accurate value for the carbon sequestration and oxygen release of the plantation forests.

The conversion factor continuous function method is more suitable for the estimation method of a regression model at the stand level, and it can better utilize the storage volume per unit area of *PM* plantations from the third-phase forest inventory data of the Guangxi Paiyangshan Forest Farm [16,18–20]; the biomass and net productivity based on the storage volume were then used to obtain the value for carbon sequestration and oxygen release in *PM* plantations:

$$A_j = 0.52V_j \tag{1}$$

$$Y_j = 5.565 \times A_j^{0.157}, \ r = 0.47, n = 15(P < 0.1) \tag{2}$$

$$M_j = Y_j \times B_j \times (1.63 \times P_c + 1.19 \times P_0) \tag{3}$$

where $A_j$ is the biomass of the aboveground forest trees per hectare in the *j*th small class of *PM* plantations (t·hm$^{-2}$); $V_j$ is the storage volume of the aboveground forest trees per hectare in the *j*th small class (t·hm$^{-2}$); $Y_j$ is the net productivity of the aboveground forest trees per hectare in the *j*th small class (t·hm$^{-2}$); $M_j$ is the total value of the carbon sequestration and oxygen release in the *j*th small class (RMB·a·$^{-1}$); $B_j$ is the total area of the *j*th small class (hm$^{-2}$); $P_c$ is the price of carbon sequestration on the market (RMB·t$^{-1}$); the Swedish carbon tax rate of 150 USD· t$^{-1}$ was used in this study (the conversion of the USD to the RMB was calculated at an exchange rate of 6.4085, which is equivalent to 961.275 RMB·t$^{-1}$); and $P_0$ is the cost of industrial oxygen production on the market (100 RMB·t$^{-1}$).

### 3.3. Soil Conservation Assessment

Soil erosion has been extensively studied both at home and abroad [21–24], and the USLE, a general-purpose soil loss equation developed by the US Department of Agriculture, has been widely used to make more accurate soil erosion estimates. This method has been widely used in the fields of evaluating soil erosion risk, watershed management, and planning as well as simulating soil and water conservation programs. According to this model, the resistance of forests to soil erosion can be introduced to understand the great utility made for soil conservation [25].

In this study, the rainfall factor, soil erosion factor, topography factor, different *PM* plantation age groups, ground cover factor, and soil as well as water conservation measures factor were considered comprehensively, and the soil carbon content as well as soil effective nitrogen, effective phosphorus, and effective potassium contents measured in the field, supplemented by reviewing the literature, were used to estimate the soil conservation values in the northern tropical case study area using the deformation formula of the *USLE*. Among them, the 0–20 cm soil layer is the main enrichment layer of soil organic carbon, and deeper soil layers are more deprived of organic carbon [26]. Therefore, this study focused on the soil organic matter of the soil layer at 0–20 cm. Within a certain range, the smaller the soil erosion rate, the better the soil function of the area, which is calculated as follows:

$$USLE_x = R_x \times K_x \times LS_x \times C_x \times P_x \tag{4}$$

$$T_h = R_x \times K_x \times LS_x \times (1 - C_x \times P_x) \tag{5}$$

$$V_a = T_h \times \sum C_i \times P_i \tag{6}$$

where $USLE_x$ denotes the soil erosion of raster x (t·hm$^{-2}$·a$^{-1}$); $R_x$ is the rainfall erosion force factor (MJ·mm·hm$^{-2}$·h$^{-1}$·a$^{-1}$); $K_x$ is the soil erodibility factor (dimensionless); $LS_x$ is the slope and slope length factor; $C_x$ is the vegetation cover factor (dimensionless); $P_x$ is the management factor (dimensionless); $T_h$ is the amount of potentially conserved soil for a single image element (t·hm$^{-2}$·a$^{-1}$); $V_a$ is the maintenance of the soil nutrient value (RMB·hm$^{-2}$·a$^{-1}$); $i$ is the type of nutrient in the soil; $C_i$ is the content of nutrient type $i$ in the soil (t·t$^{-1}$); and $P_i$ is the market price of nutrient type $i$ (RMB·t$^{-1}$). The soil conservation value of the planted pine forest ecosystem was estimated based on the average market price of fertilizers, where diammonium phosphate contained 14.0% nitrogen and 15.01% phosphorus, while potassium chloride contained 50.0% potassium; the price of diammonium phosphate fertilizer was 2400 RMB·t$^{-1}$, potassium chloride fertilizer was 2200 RMB·t$^{-1}$, and organic matter was 320 RMB·t$^{-1}$.

- Rainfall Erosion Force Factor, *R*

Considering the difficulty of obtaining multiyear daily rainfall data in the study area, the empirical formula of Wischmeier [27] was used in this study to calculate the rainfall erosion force factor with monthly rainfall base data. The formula is as follows:

$$R = M \times 17.02 = \sum_{i=1}^{12} 1.735 \times 10^{\left(1.5 \times log_{10}\left(\left(P_i^2 / P\right) - 0.8188\right)\right)} \times 17.02 \tag{7}$$

where $R$ is the rainfall erosion force factor; $P_i$ is the $i$th month rainfall; and $P$ is the annual rainfall. The unit of $M$ is the American system unit, and the conversion into the international system unit (MJ·mm·hm$^{-2}$·h$^{-1}$·a$^{-1}$) needs to be multiplied by 17.02 when used.

- Soil Erodibility Factor, *K*

Soil erodibility is an important aspect of soil properties and is a performance evaluation of the susceptibility of soil to damage by erosion camp forces, as well as the sensitivity of soil to the effects of erosion camp force separation and transport. The index of the soil erodibility factor, *K* [28], is commonly used internationally to measure the ability of soil itself to resist impact, and the magnitude of the *K* value indicates how difficult it is for soil

to be eroded. Y.L. et al. [29] calculated the soil erodibility factor, *K,* for each soil type in the eastern hilly region of China based on second soil census data, starting with soil subclasses; the soil types in the study area of this study and the corresponding *K* values were taken as 0.214 for the ruddy loam *K* value and 0.231 for that of the red loam.

- Slope and Slope Length Factor, *LS* [30,31]

The topographic factor is the condition under which soil erosion occurs; it influences the redistribution of rainfall as well as external camp forces, and has an important role in the process of soil erosion [32]. The slope and slope length factor are roughly positively correlated with soil erosion [33], and, generally, the greater the slope the greater the kinetic energy after rainfall confluence and the more severe the soil erosion caused; the longer the slope length the longer the rainfall runoff confluence time and the greater the soil erosion caused.

In this study, the *USLE* general-purpose soil erosion model, which is more accurate in practical calculations, was used to carry out an evaluation of soil erosion in the area of the forest-scale *PM* plantation forests, and a spatial analysis needed to be carried out with the help of a digital elevation model (DEM), which is calculated as follows:

$$\begin{cases} S = 10.80 \times \sin\theta + 0.03 & \theta < 5° \\ S = 16.80 \times \sin\theta - 0.50 & 5° \leq \theta < 10° \\ S = 21.91 \times \sin\theta - 0.96 & \theta \geq 10° \end{cases} \tag{8}$$

$$L = \left( {}^{\alpha}/_{22.13} \right)^{m} \tag{9}$$

$$\begin{cases} m = 0.5 & \tan\theta > 0.05 \\ m = 0.4 & 0.03 < \tan\theta \leq 0.05 \\ m = 0.3 & 0.01 < \tan\theta \leq 0.03 \\ m = 0.2 & \tan\theta \leq 0.01 \end{cases} \tag{10}$$

where *S* is the slope factor (dimensionless); *L* is the slope length factor (dimensionless); $\theta$ is the slope (°); $\alpha$ is the slope length (m); *m* is the slope length index (dimensionless); and 22.13 is the standard plot slope length of 22.13 m.

In the actual survey process, it is difficult to obtain the exact value of the slope length of each grid point, and some scholars have found that the slope length becomes larger as the slope becomes smaller [22]. According to field survey regulations [34], at less than 10° the slope length is set at 60 m; between 10° and 15° the slope length is set at 50 m; between 15° and 20° the slope length is set at 40 m; between 20° and 25 ° the slope length is set at 30 m; between 25° and 30° the slope length is set at 20 m; and above 35° the slope length is set at 15 m, where the slope interval value follows the principle of the front closed and then open.

- Vegetation and Management Factor, *C*

In regional soil erosion surveys the ground cover factor is one of the most important determination factors [35], and the vegetation of *PM* plantations has the effect of slowing down soil erosion and conserving soil. According to observations, forests can reduce surface runoff and soil erosion by more than 70% on average [36], and forestry as well as grass measures are important and effective biological management measures. The interception effect of vegetation on rainfall often varies depending on the difference in vegetation cover, and the greater the vegetation cover the better the interception effect.

In this study the vegetation cover index was calculated using ENVI software, based on Landsat image data in the case study area:

$$c = (NDVI - NDVI_{min}) \Big/ (NDVI_{max} - NDVI_{min}) \tag{11}$$

where $c$ is the vegetation cover (%), *NDVI* is the normalized vegetation index value, and $NDVI_{max}$ and $NDVI_{min}$ are the maximum and minimum values of the *NDVI* in the study area, respectively.

Applying the current and more widespread method of calculating vegetation management factors based on vegetation cover [37,38], the calculation formula is as follows:

$$C = \begin{cases} 1 & c \leq 1\% \\ 0.6508 - 0.3436 \times \log_{10} c & 1\% < c \leq 78.3\% \\ 0 & c > 78.3\% \end{cases} \tag{12}$$

where $c$ is the vegetation cover. The vegetation cover of *PM* plantations basically increases with the age of a *PM* forest, and the vegetation cover can reflect the superiority or inferiority of the vegetation and management factor, *C*.

- Soil Conservation Measures Factor, *P*

At present, there is no unified standard for assigning the *P* value of the soil and water conservation factor in China. In this study, the soil and water conservation factor is the ratio of the soil loss under specific soil and water conservation measures to the soil loss under undulating tillage, with a *P* between 0 and 1 [39]. The *P* values corresponding to different slopes under the contour strip tillage model were chosen [25]: the *P* value for slopes less than 5° was taken as 0.3, the *P* value for slopes between 5° and 10° was taken as 0.5, and for slopes greater than 10°, a *P* value of 0.6 was taken.

*3.4. Water Conservation*

The water production model in the InVEST model implements a quantitative, dynamic, and visual evaluation of the ecosystem water production functions. It is based on the Budyko curve and water balance principles to quantitatively analyze the abstract ES functions at a uniform pixel scale. The actual evapotranspiration was subtracted from the precipitation at each pixel to estimate the water production of the pixel, including the surface runoff, soil water content, apoplastic water holding capacity, and canopy interception [40].

The water yield model of the InVEST model in this study was mainly based on the annual rainfall and Budyko curve, taking into account the data of the land-use type (forest age type), rainfall, evapotranspiration, etc. The calculation method is as follows:

$$Y_{xj} = \left(1 - AET_{xj} \Big/ P_x \right) \cdot P_x \tag{13}$$

$$AET_{xj} \Big/ P_x = 1 + PET_{xj} \Big/ P_x - \left[1 + \left(PET_{xj} \Big/ P_x\right)^{\omega}\right]^{1/\omega} \tag{14}$$

$$PET_{xj} = K_{cx} \times ET0_x \tag{15}$$

$$\omega_x = \left(AWC_x \times Z\right) \Big/ P_x + 1.25 \tag{16}$$

$$AWC_x = min(MaxSoilDepth_x, RootDepth_x) \times PAWC_x \tag{17}$$

where $Y_{xj}$ is the annual water yield (mm) of image $x$ in forest age type $j$ and $AET_{xj}$ is the actual annual evapotranspiration (mm) of image $x$ in forest age type $j$. The evapotranspiration of the different plant community types varies significantly and needs to be considered on a case-by-case basis [41]. $P_x$ is the annual rainfall (mm) of image x; $AET_{xj}/P_x$ is an approximation of the Budyko curve; $\omega$ is a nonphysical parameter indicating the ratio of the potential evapotranspiration to the precipitation; $K_c$ is the plant evapotranspiration coefficient; *ET0* is the potential evapotranspiration; *AWC* indicates the effective soil water content; and *AWC* indicates the plant's available water content. MaxSoilDepth is the maximum soil depth, which can be obtained from the data on the forest secondary resource

survey, and RootDepth is the depth of the root system. The specific data and calculation of each parameter are shown below.

- Potential Evapotranspiration

The potential evapotranspiration emission, *PET*, has the same meaning as the crop reference evapotranspiration, *ET0*, in the model, which is the amount of water spilled through soil evaporation and plant evapotranspiration in millimeters. Due to the difficulty of obtaining data such as the soil heat flux density and saturated water pressure, the Modified-Hm2 Hargreaves method equation was used in this study, which is calculated as follows:

$$ET0 = 0.0013 \times 0.408 \times RA \times (Tavg + 17) \times (TD - 0.0123P)^{0.76} \tag{18}$$

where *ET0* is the potential evapotranspiration; *RA* represents the solar top atmospheric radiation; *Tavg* is the mean value of the maximum and minimum temperature (°C) of the forest site in the study area; *TD* is the interpolated value of the mean maximum and mean minimum temperature (°C) of the forest site; and *p* is the rainfall (mm). The data required for this equation, such as the solar top atmospheric radiation, were obtained from the solar radiation data purchased from national weather stations.

- Land-Use Map

The vector map of the different age stands of the *PM* plantations was extracted from the vector map of the forest site and converted into raster data in ArcGIS with a resolution of 30 × 30 m.

- Water Content Available to Plants

For the proportion of the water in the soil that can be absorbed or utilized by plants, this paper used the nonlinear fitting soil *PAWC* model of W.Z. et al.'s [42] nonlinear fitted soil *PAWC* model for estimating a plant's available water content, as per the following equation:

$$PAWC = 54.509 - 0.132 \times SAND - 0.003 \times SAND^2 - 0.055 \times SILT - \\ 0.006 \times SILT^2 - 0.738 \times CLAY + 0.007 \times CLAY^2 - 2.688 \times C + 0.501 \times C^2 \tag{19}$$

where *SAND* is the percentage of the content of gravel (%); *SILT* is the percentage of the content of powder (%); *CLAY* is the percentage of the content of clay (%); and *C* is the percentage of the content of organic matter (%).

- Table of Biophysical Parameters (Table 3)

**Table 3.** Table of the biophysical parameters of *PM* plantations in the northern tropical case study area.

| Age Group | Lucode | Kc | Root_Depth | LULC_Veg |
|---|---|---|---|---|
| Young forest | 2 | 0.96 | 600 | 1 |
| Mid-aged forest | 3 | 0.96 | 600 | 1 |
| Mature forest | 4 | 0.96 | 600 | 1 |
| Over-mature forest | 5 | 0.96 | 1000 | 1 |
| Other | 1 | 0.5 | 1000 | 0 |

- Z

Z is an empirical coefficient that characterizes the seasonal distribution of precipitation and hydrogeological features. Donohue [43] found that the *Z* coefficient was positively related to the amount of precipitation, and combined with J.R.'s [44] results on the empirical *Z* coefficient of the water yield of the ecosystem of the Xijiang River Basin in Guangxi, as well as the precipitation characteristics of this study area, the empirical coefficient of *Z* was taken as 10 in this study.

To obtain the annual rainfall and evapotranspiration in the study area in 2009, 2013, and 2018, this study mainly collected and purchased meteorological data from 40 meteorological stations near Guangxi and spatially interpolated the annual rainfall data using the spatial kriging interpolation method in ArcGIS. After deriving the material quality according to the InVEST model, the shadow value method was used to estimate the value for the water conservation in the *PM* plantation forests, and the cost of constructing a reservoir was used to estimate the value of regulating the water quantity in the *PM* plantation forest ecosystem. The cost of the unit reservoir capacity was 6.11 RMB·m$^{-3}$, and the cost of the water purification was used to estimate the value of the purifying water quality of *PM* plantation forests, which was integrated as the value of the water connotation.

*3.5. Ecosystem Functional Zoning Method for PM Plantation Forests*

ES clustering is an effective method for conducting the spatial partitioning of ESs to assist managers in achieving multi-ecosystem service management [45], specifically the study of the partitioning of a series of spatially as well as temporally recurring ESs [46] and then adopting different management strategies for different service partitions.

In this study, forestry small classes were used as ecological service cluster units, and the boundaries of small class units were not broken when partitioning. The differences between adjacent or close small class units were different, which required the further use of a cluster analysis based on data mining and pattern recognition to divide the samples into different categories with the smallest possible internal variability in the region and the largest possible external variability [47]. This partitioning method can be abstracted into four steps: feature selection, algorithm selection, validation of validity, and interpretation of results [46].

In this study, we took the small class of the *PM* plantation forests in the northern tropical case study area in 2018 as the basic unit, and the value of the ESs per unit area of the small class of four categories as the data source. The spectral clustering method in geoda software was used to obtain the service cluster division area. Spectral clustering is an algorithm developed from graph theory, with strong adaptability to data distribution and an excellent clustering effect, which is subsequently widely used in clustering [48–50]. Its advantage is that it considers all data as points in space and takes into account the spatial distance effect. Because of the principle of area dominance, the principle of regional integrity and regional conjugacy, and the ease of supervision and management that should be followed in the ecological geospatial zoning process, small groups of *PM* plantation forests were gradually merged with the editor and merging tools in ArcGIS 10.5 software on the basis of the results of preferential clustering.

## 4. Results

*4.1. Spatial Distribution of the ES Values in PM Plantation Forests*

4.1.1. Value of Wood Supply

The spatial distribution of the multiyear timber supply value in *PM* plantations in the northern tropical case study area is shown in Figure 2. In 2009, the total value of the timber provided was RMB 60,831,073, and the value of the timber provided per hectare was RMB 7587.19/hm$^2$, with significant spatial variability. In 2013, the total value of the timber provided was RMB 53,295,493, and the value of the timber provided per hectare was RMB 8527.55/hm$^2$. In 2018, the total value of the timber provided was RMB 755,415,591, and the value of the timber provided per hectare was RMB 12,559.08/hm$^2$. The total value of the timber provided in the northern tropical case study area decreased and then increased from 2009 to 2018 because the area of *PM* plantations decreased by 1768 hm$^2$ in 2013, while the area decreased after 5 years; however, with the growth of *PM* plantations and the increase in the stocking volume, its timber provision capacity also increased.

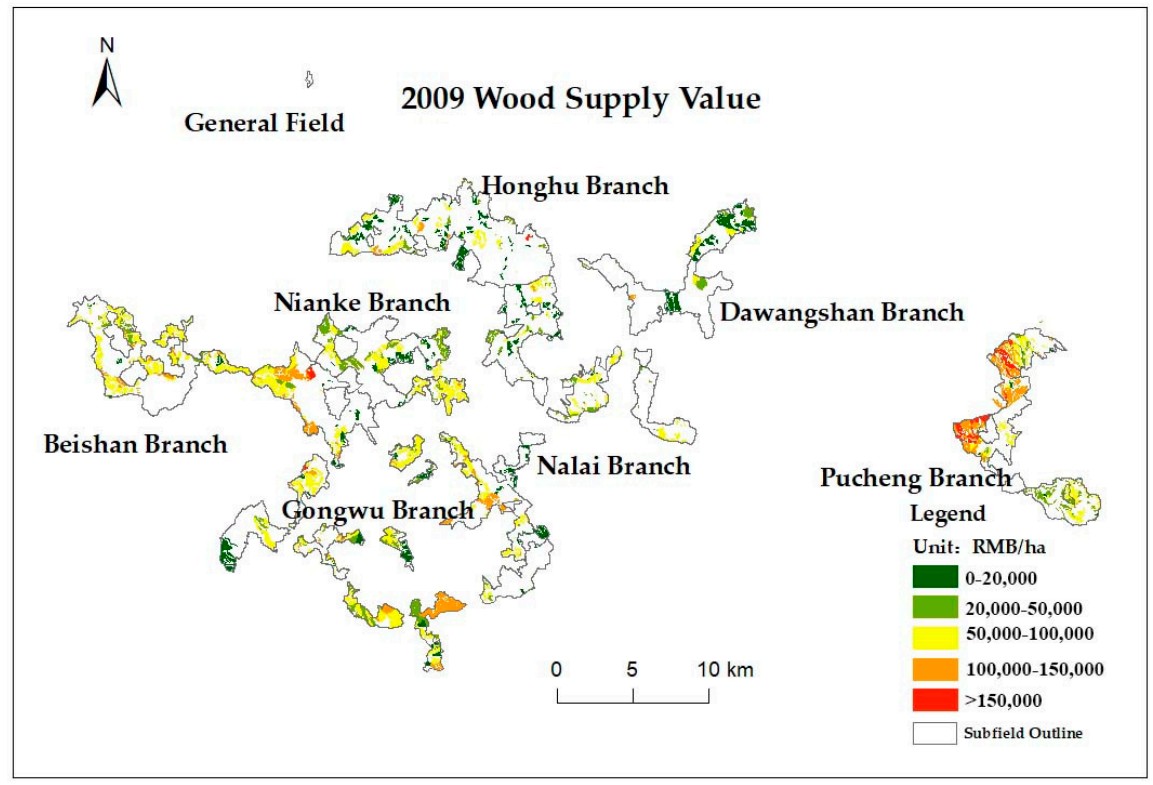

(**a**)

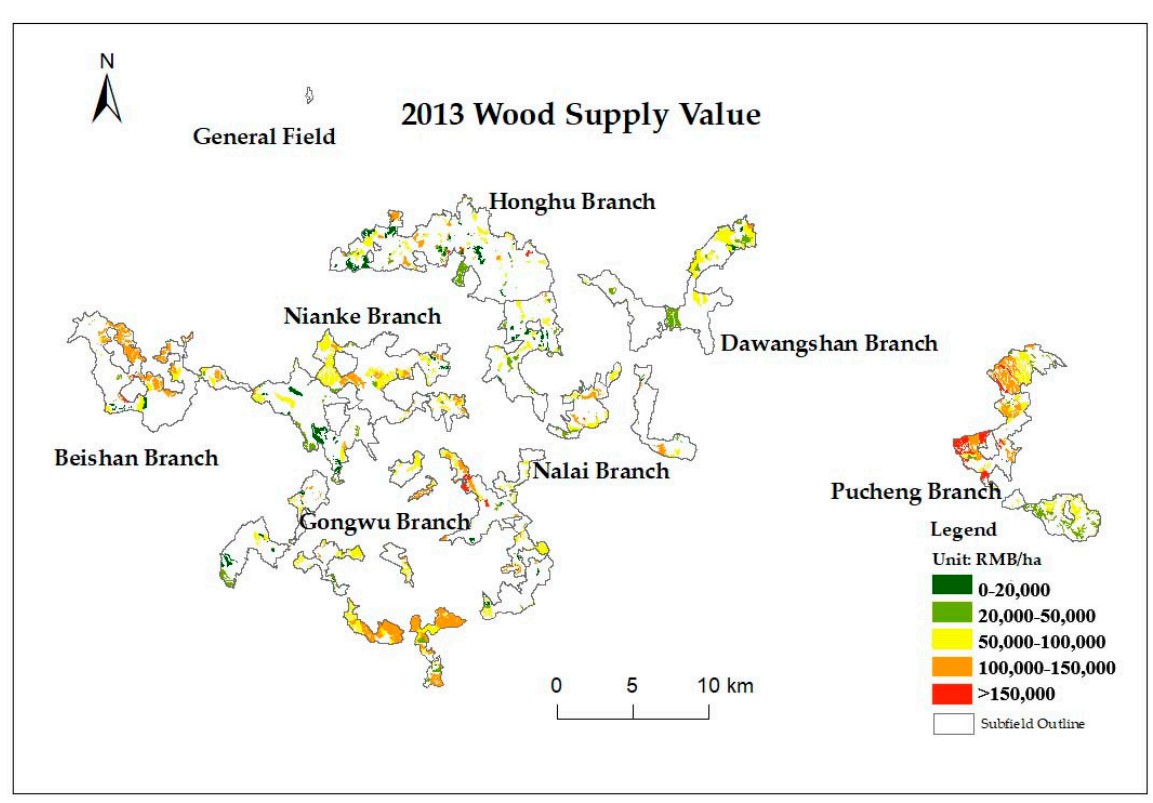

(**b**)

**Figure 2.** *Cont.*

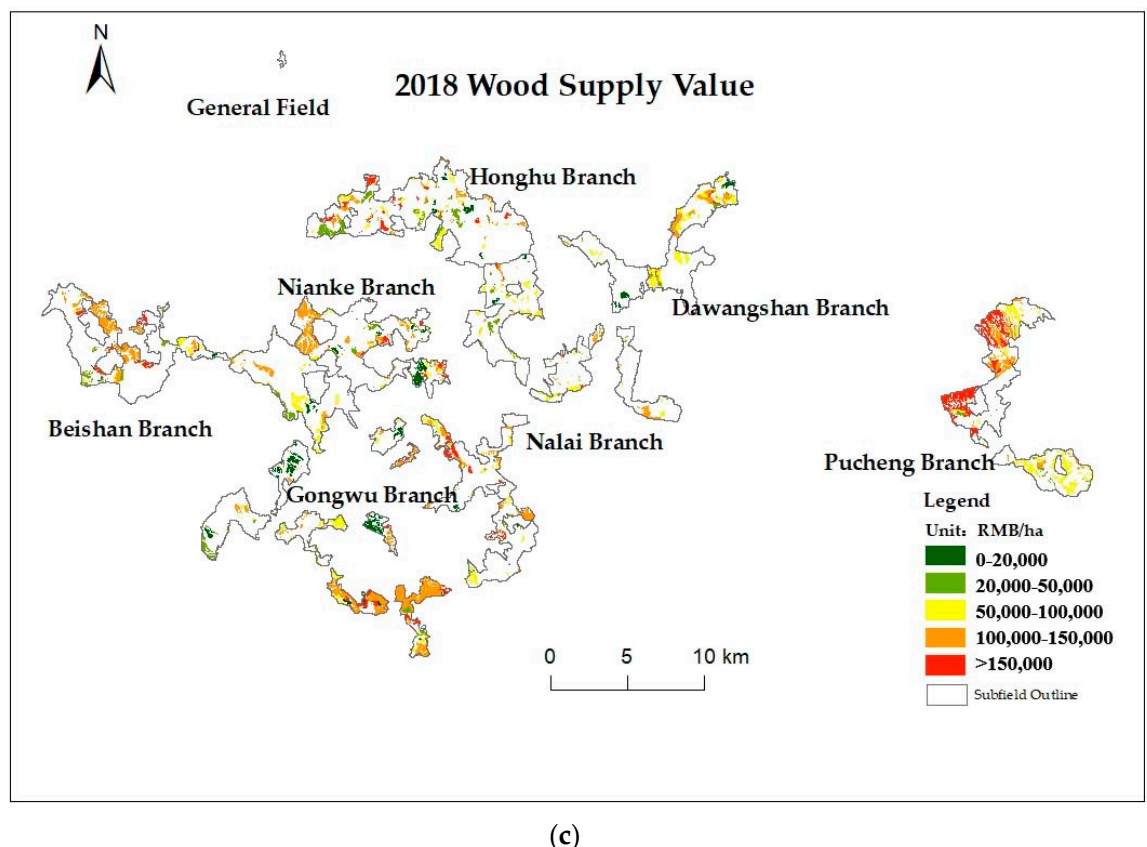

**(c)**

**Figure 2.** Spatial distribution map of the wood supply value for *PM* plantations in the northern tropical case study area in 2009–2018. (**a**) 2008 Wood supply value; (**b**) 2013 Wood supply value; (**c**) Wood supply value.

4.1.2. Value of Carbon Sequestration and Oxygen Release

The spatial distribution of the multiyear sequestration and oxygen release value in the northern tropical case study area of the *PM* plantations is shown in Figure 3. In 2009, the total value for the carbon sequestration and oxygen release was RMB 196,834,390, and the value of the carbon sequestration and oxygen release per hectare was 24,550.29 RMB/hm$^2$, with significant spatial variability. In 2013, the total value of carbon sequestration and oxygen release was RMB 170,686,939, and the value for the carbon sequestration and oxygen release per hectare was 27,310.78 RMB/hm$^2$. In 2018, the total value of carbon sequestration and oxygen release was RMB 165,296,497, and the value of the carbon sequestration and oxygen release per hectare was 27,481.17 RMB/hm$^2$. The spatial distribution characteristics of the sequestered carbon and oxygen release value were influenced by the net productivity of *PM* plantation forests; the higher the stockpile the higher the biomass of *PM* plantation forest small groups and the higher their net productivity, which is also the reason why the spatial distribution pattern of the sequestered carbon and oxygen release value was similar to that of the timber-provided value map. From 2009 to 2018, the value of the carbon sequestration and oxygen release of *PM* plantations in the northern tropical case study area decreased continuously, and the value of the carbon sequestration and oxygen release per hectare increased continuously, which was related to the decrease in the area of *PM* plantations and the natural growth of forest stands.

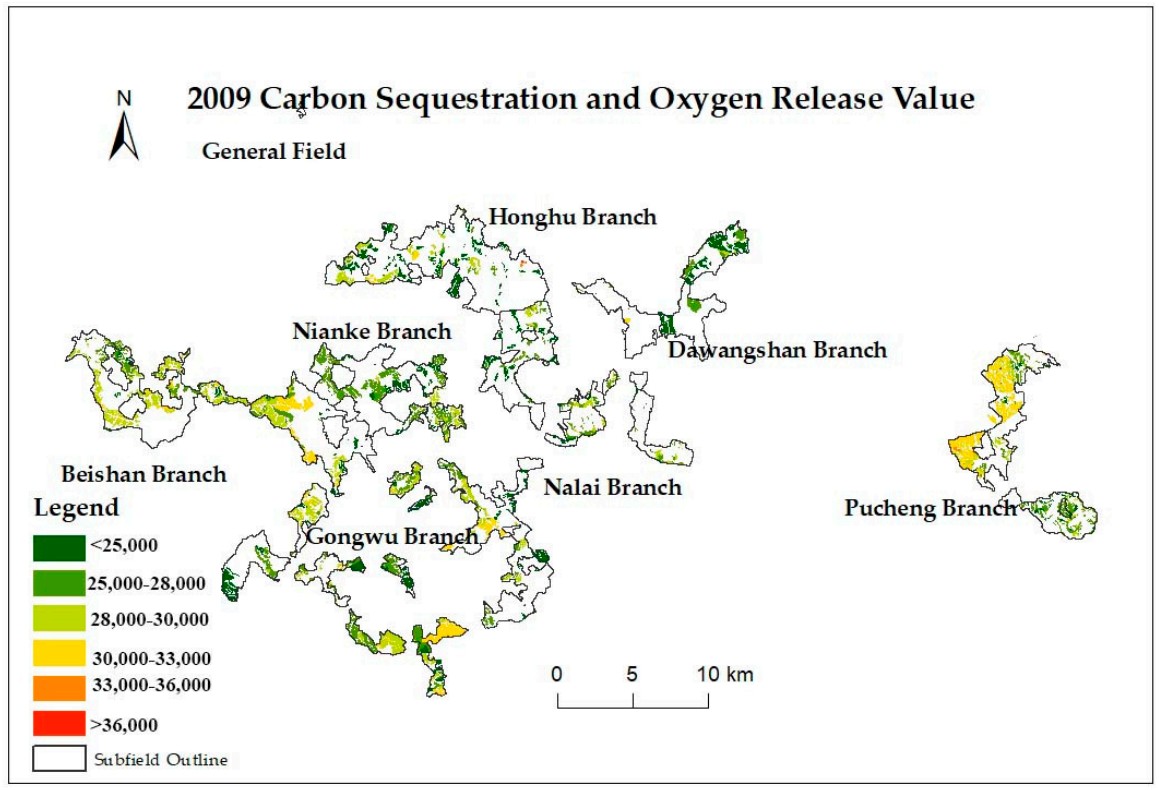

(**a**)

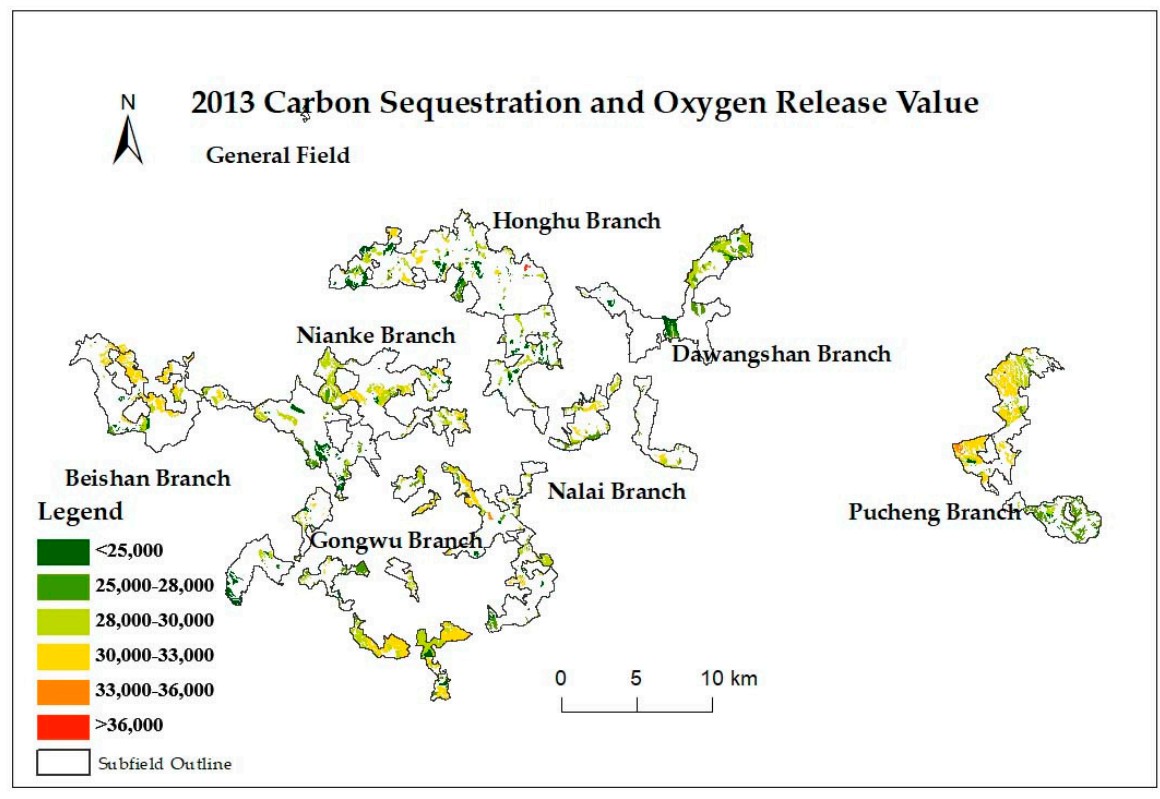

(**b**)

**Figure 3.** *Cont*.

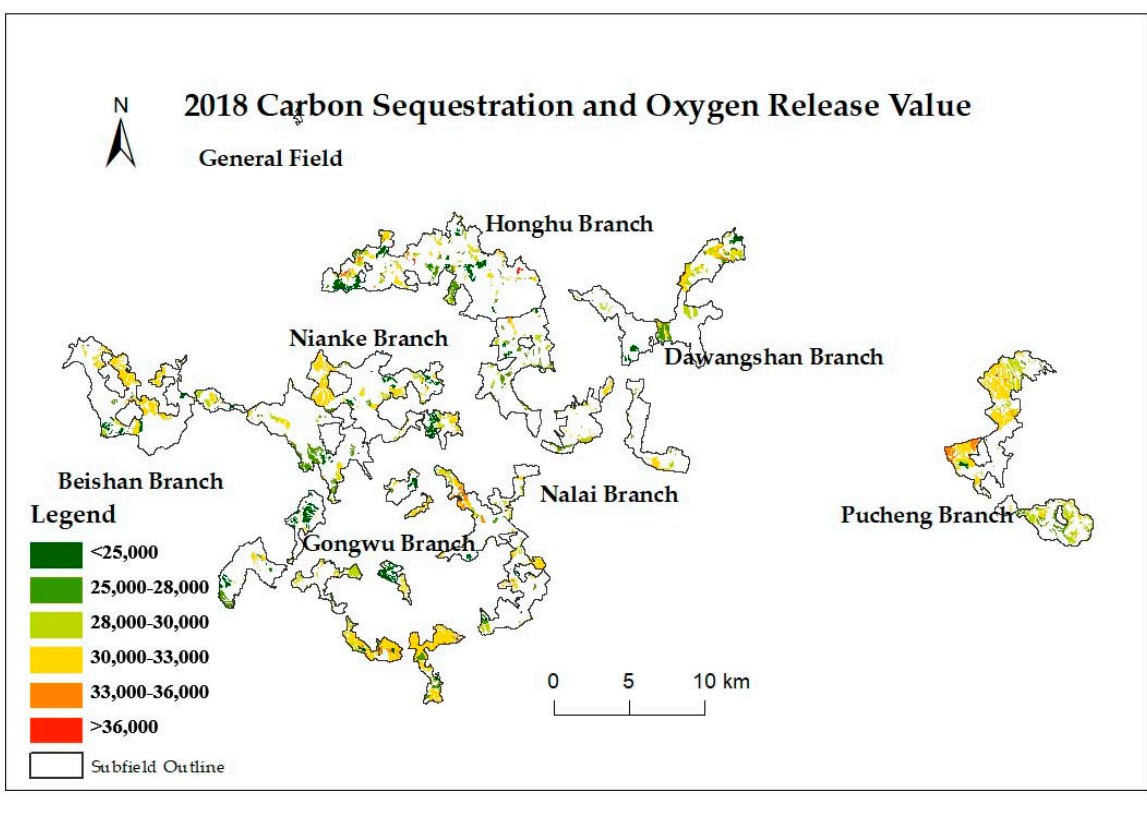

(**c**)

**Figure 3.** Spatial distribution map of the carbon sequestration and oxygen release values of *PM* plantations in the northern tropical case study area in 2009–2018. (**a**) 2008 Carbon sequestration and oxygen release value; (**b**) 2013 Carbon sequestration and oxygen release value; (**c**) 2018 Carbon sequestration and oxygen release value.

4.1.3. Value of Soil Conservation

The spatial distribution of the multiyear soil conservation value of the *PM* plantation forests in the northern tropical case study area is shown in Figure 4. In 2009, the total soil conservation value was RMB 3,267,240, and the soil conservation value per hectare was 4075.04 RMB/hm$^2$, with obvious spatial variability. In 2013, the total soil conservation value was RMB 29,152,885, and the soil conservation value per hectare was 4664.61 RMB/hm$^2$, with obvious spatial heterogeneity in the overall soil conservation value. In 2018, the total soil conservation value was RMB 37,271,819, and the soil conservation value per hectare was 6081.52 RMB/hm$^2$. The overall capacity of the soil conservation value provision increased. From 2009 to 2018, the soil conservation value in the northern tropical case study area showed a trend of increasing and then decreasing, and the soil conservation value per hectare increased. In 2013, the total value of the soil conservation decreased and the value per hectare increased because the total area decreased but the level of the soil conservation per unit area increased.

4.1.4. Value of Water Conservation

The spatial distribution of the multiyear water conservation value of *PM* plantations in the northern tropical case study area is shown in Figure 5. The spatial distribution of the water conservation value of *PM* plantation forests in the northern tropical case study area from 2009 to 2018 showed a trend of being low in the northwest and high in the southeast, and the total value as well as the water conservation value per hectare both showed an increasing and then decreasing trend, mainly because the rainfall in the northern tropical case study area in 2013 was much larger than that in 2009 and 2018.

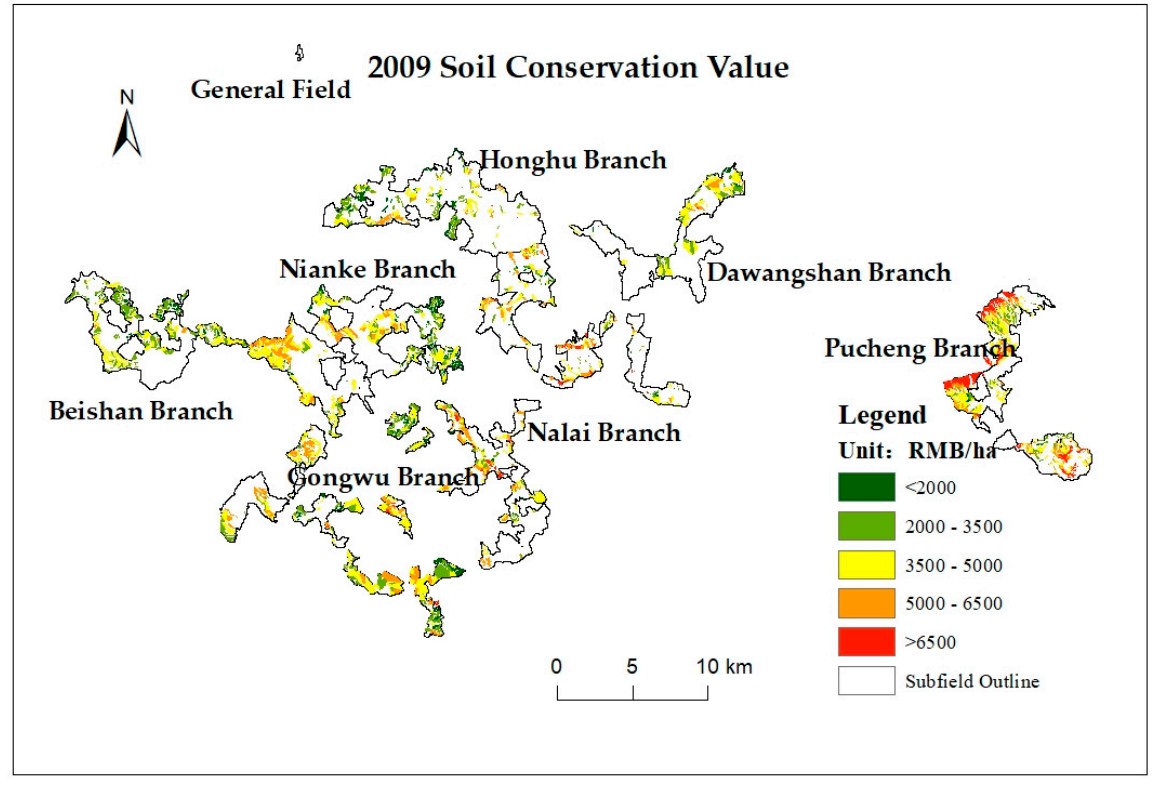

(**a**)

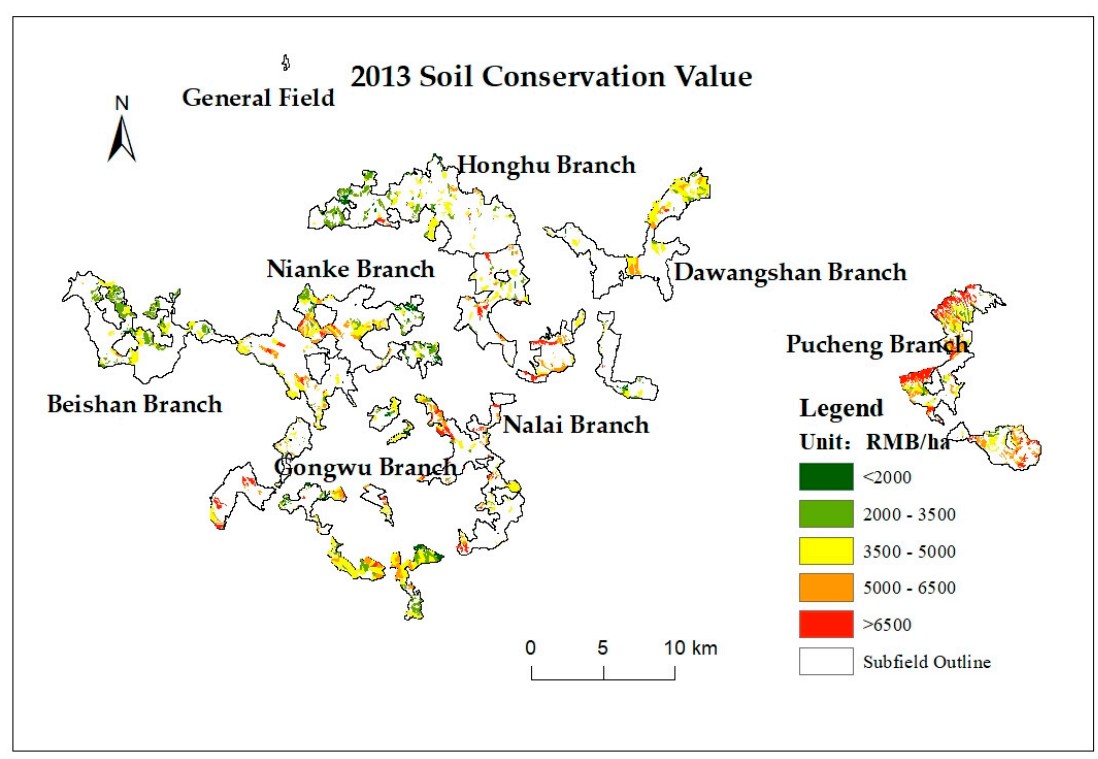

(**b**)

**Figure 4.** *Cont.*

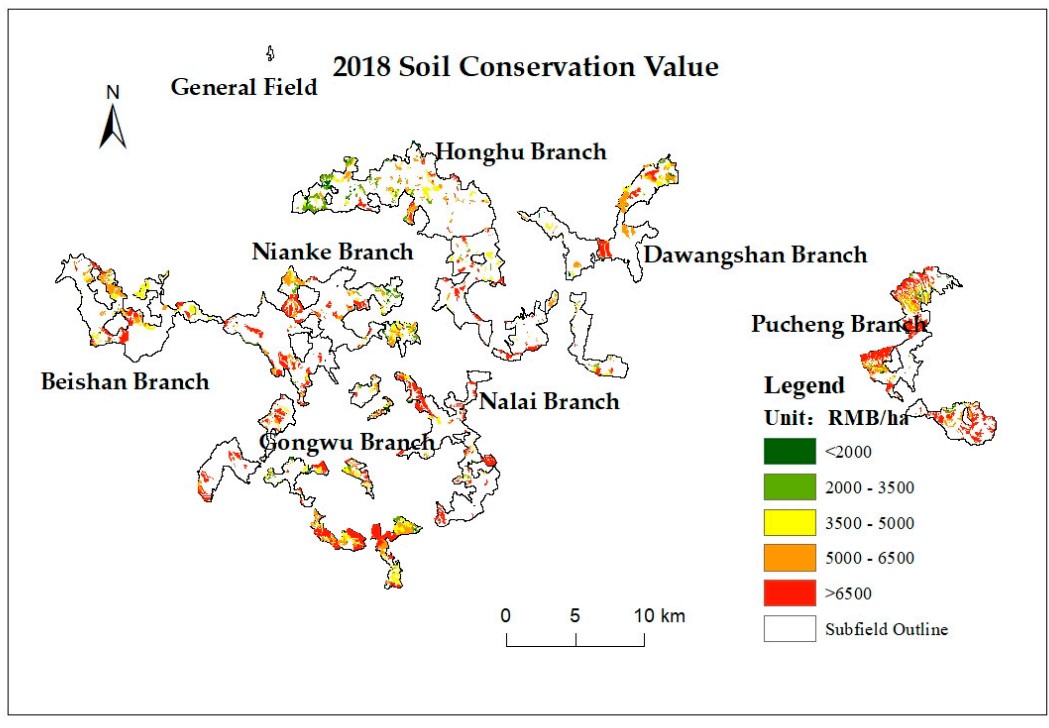

(**c**)

**Figure 4.** Spatial distribution of the soil conservation values of *PM* plantations of the northern tropical case study area in 2009–2018. (**a**) 2008 Soil conservation value; (**b**) 2013 Soil conservation value; (**c**) 2018 Soil conservation value.

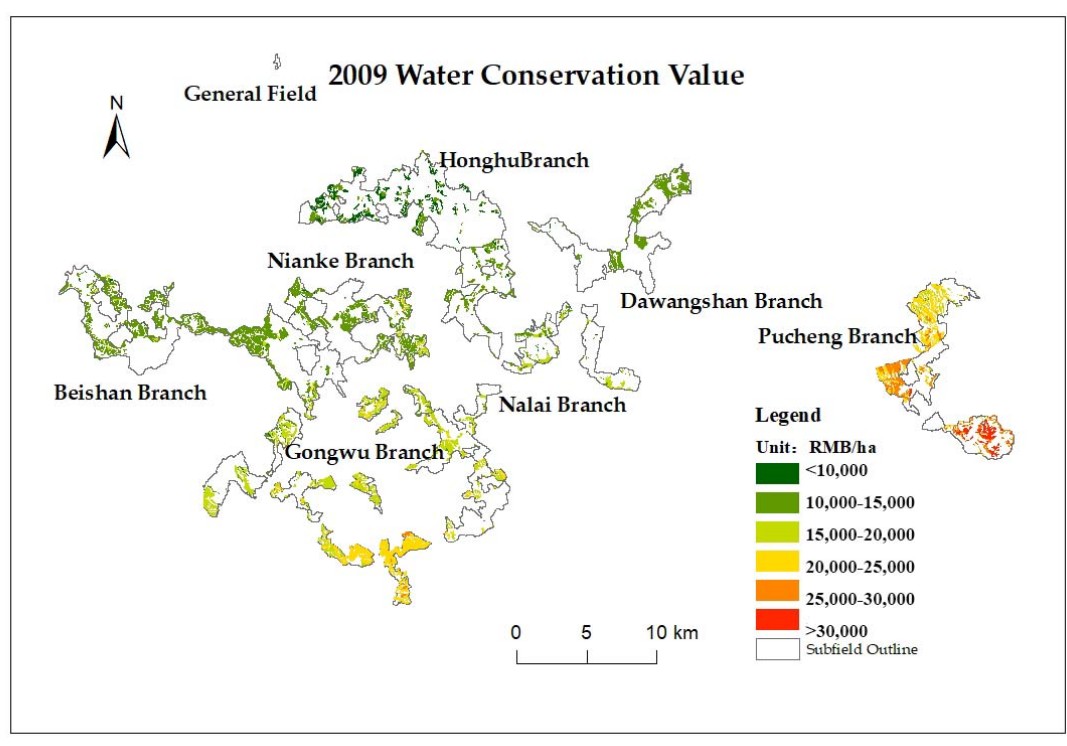

(**a**)

**Figure 5.** *Cont.*



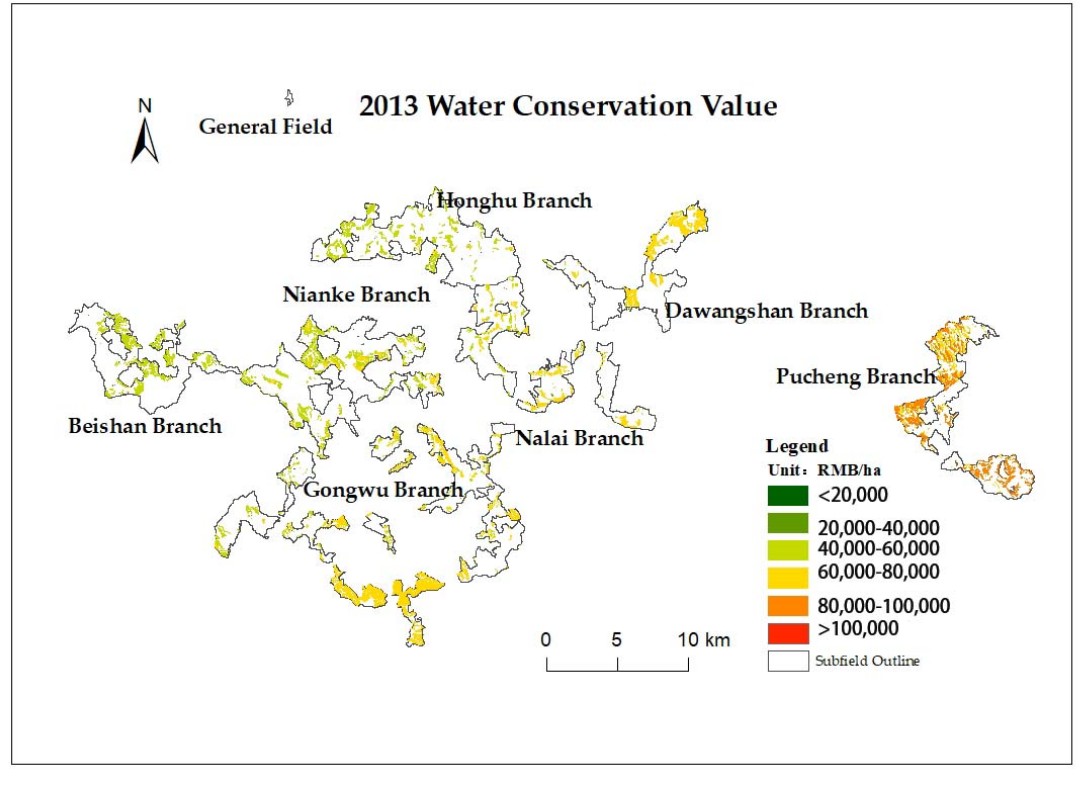

(b)

(c)

**Figure 5.** Spatial distribution map of the water conservation values of *PM* plantations in the northern tropical case study area in 2009–2018. (**a**) 2008 Water conservation value; (**b**) 2013 Water conservation value; (**c**) 2018 Water conservation value.

### 4.2. Total Value of the ES Functions of PM Plantation Forests

Table 4 reflects the information regarding the total value and value per hectare of the four ES functions for the different age groups of *PM* plantations in the northern tropical case study area from 2009 to 2018.

**Table 4.** Value of the four ESs of *PM* plantations in the differently aged forests in the northern tropical case study area in 2009–2018.

| Age Group | | Total Value (RMB) | | | Value per Hectare (RMB/hm$^2$) | | |
|---|---|---|---|---|---|---|---|
| | | **2009** | **2013** | **2018** | **2009** | **2013** | **2018** |
| Wood provided | Young forest | 3,626,545.59 | 5,053,202.81 | 1,163,518.67 | 1932.61 | 3186.73 | 1379.72 |
| | Mid-aged forest | 41,724,764.74 | 20,332,615.24 | 26,474,930.91 | 9012.80 | 10,014.59 | 13,442.46 |
| | Mature forest | 14,593,240.77 | 27,134,389.43 | 47,280,788.44 | 9702.95 | 10,339.27 | 14,809.03 |
| | Over-mature forest | 886,522.89 | 775,285.02 | 622,353.24 | 116,647.75 | 82,477.13 | 66,207.79 |
| Carbon sequestration and oxygen release | Young forest | 22,813,913.80 | 33,349,000.29 | 9,824,829.77 | 12,157.69 | 21,031.09 | 11,650.46 |
| | Mid-aged forest | 131,458,104.55 | 59,769,116.69 | 57,704,511.40 | 28,395.75 | 29,438.56 | 29,299.07 |
| | Mature forest | 42,290,373.19 | 77,206,545.73 | 97,419,505.92 | 28,118.60 | 29,418.74 | 30,513.20 |
| | Over-mature forest | 271,998.92 | 362,276.57 | 347,650.23 | 35,789.33 | 38,540.06 | 36,984.07 |
| Soil conservation | Young forest | 6,931,672.81 | 6,707,137.35 | 4,084,525.81 | 3693.94 | 4229.76 | 4767.74 |
| | Mid-aged forest | 19,030,434.89 | 9,966,828.18 | 12,192,438.87 | 4110.69 | 4909.04 | 6164.96 |
| | Mature forest | 6,685,359.42 | 12,451,522.81 | 20,955,551.11 | 4445.05 | 4744.52 | 6379.36 |
| | Over-mature forest | 24,572.51 | 27,396.34 | 39,303.06 | 3233.23 | 2914.50 | 4181.18 |
| Water conservation | Young forest | 26,326,176.13 | 93,916,187.05 | 29,167,253.37 | 14,029.40 | 59,226.96 | 34,046.05 |
| | Mid-aged forest | 73,858,126.65 | 128,296,068.06 | 66,509,996.62 | 15,953.80 | 63,190.69 | 33,629.97 |
| | Mature forest | 31,228,574.57 | 179,911,098.91 | 132,917,785.03 | 20,763.68 | 68,553.23 | 40,463.27 |
| | Over-mature forest | 79,051.59 | 536,922.88 | 250,274.77 | 10,401.53 | 57,119.45 | 26,624.98 |

Firstly, in terms of the total value provided by the wood, the mid-aged forest provided the largest total value in 2009, the mature forest provided the largest total value in 2013 and 2018, and the over-mature forest provided the smallest total value throughout the years, mainly because the over-mature forest exists in a small area, but its value provided per hectare of wood was much larger than that of the forests of other ages. The total value of the young forests increased and then decreased over the 10-year period. The total value of the mid-aged forests decreased and then increased over the 10-year period. The total value of the mature forests kept increasing over the 10-year period, and the total value of the over-mature forests continued to decrease over the 10-year period, probably due to the occurrence of natural disasters and human felling. The value of the wood per hectare was the lowest in the young forests, ranging from 1379.72 to 3186.73 RMB/hm$^2$, and the highest in the mature forests, ranging from 66,207.79 to 116,647.75 RMB/hm$^2$. The value per hectare of the young forests increased and then decreased over the 10 year-period, while the value per hectare of the mid-aged as well as mature forests increased, while it decreased in the over-mature forests.

Secondly, in terms of the total value of the carbon sequestered and oxygen released, the mid-aged forest provided the largest total value of the carbon sequestered and oxygen released in 2009. The mature forest provided the largest total value of the carbon sequestered and oxygen released in 2013 and 2018, and the over-mature forest provided the smallest total value for multiple years. The relationship between the magnitude of the value provided by the carbon sequestered and oxygen released in the differently aged stands was similar to that provided by the wood. In terms of the value per hectare of the carbon sequestered and oxygen released, it was the largest in over-mature forests from 2009 to 2018, at 35,789.33–38,540.06 RMB/hm$^2$. The value per hectare of the carbon sequestered and oxygen released was the smallest in young forests, at 11,650.46-21,031.09 RMB/hm$^2$. The value per hectare of the carbon sequestered and oxygen released in the differently aged forests from 2009 to 2013 was, from smallest to largest, the young forests, mature forests, mid-aged forests, and over-mature forests.

Furthermore, in terms of the total value of soil conservation, the mid-aged forests provided the greatest total value in 2009 and 2013, the mature forests provided the greatest total value in 2018, and the over-mature forests provided the least total value from 2009 to 2018. The total value of the mature and over-mature forests kept increasing over the 10-year period, while the total value of the young and mid-aged forests first decreased and then

increased over the 10-year period. In terms of the value per hectare of soil conservation, the value per hectare of soil conservation was the lowest in the over-mature forests, ranging from 2914.50 to 4181.18 RMB/hm$^2$, and the value per unit area of soil conservation was the highest in mature forests, ranging from 4445.05 to 6379.36 RMB/hm$^2$. The value per unit area of the young, mid-aged, and mature forests maintained an increasing trend over the 10-year period, while the value per unit area of the over-mature forests showed a decreasing and then increasing trend over the 10-year period.

Finally, in terms of the total value of the water conservation, the mid-aged forest provided the largest total value of water conservation in 2009, the mature forest provided the largest total value in 2013 and 2018, and the over-mature forest provided the smallest total value from 2009 to 2013. In terms of the value per hectare of water conservation, the mature forests provided the largest value of water conservation per hectare in 2018, at 40,463.27 RMB/hm$^2$, the over-mature forests provided the smallest value of water conservation per hectare, at 26,624.98 RMB/hm$^2$, and the value of the water conservation per hectare of the differently aged forests over the years was, from small to large, the over-mature forests, mid-aged forests, young forests, and mature forests.

### 4.3. Results of Ecosystem Function Zoning in PM Plantation Forests

According to the four different ESs of *PM* plantation forests in the northern tropical case study area, using the spectral clustering method, and considering the spatial relationships of the ESs as well as the principle of the continuity of the service partitioning, the number of small groups contained in the ES clusters obtained from the partitioning results and the difference results of the mean value of the final clustering centers are shown in Table 5. In this study, according to the pattern characteristics of the three types of ES clusters, the *PM* plantation forest ecosystem was divided into three types of service zones, namely the wood supply zone, the ecological nourishment zone, and the ecological restoration zone. The development and utilization of the northern tropical *PM* plantation forests can focus on the wood supply zone and the ecological nourishment zone, while the ecological restoration zone accounts for a relatively small amount. The spatial distribution of each type of ES cluster is shown in Figure 6.

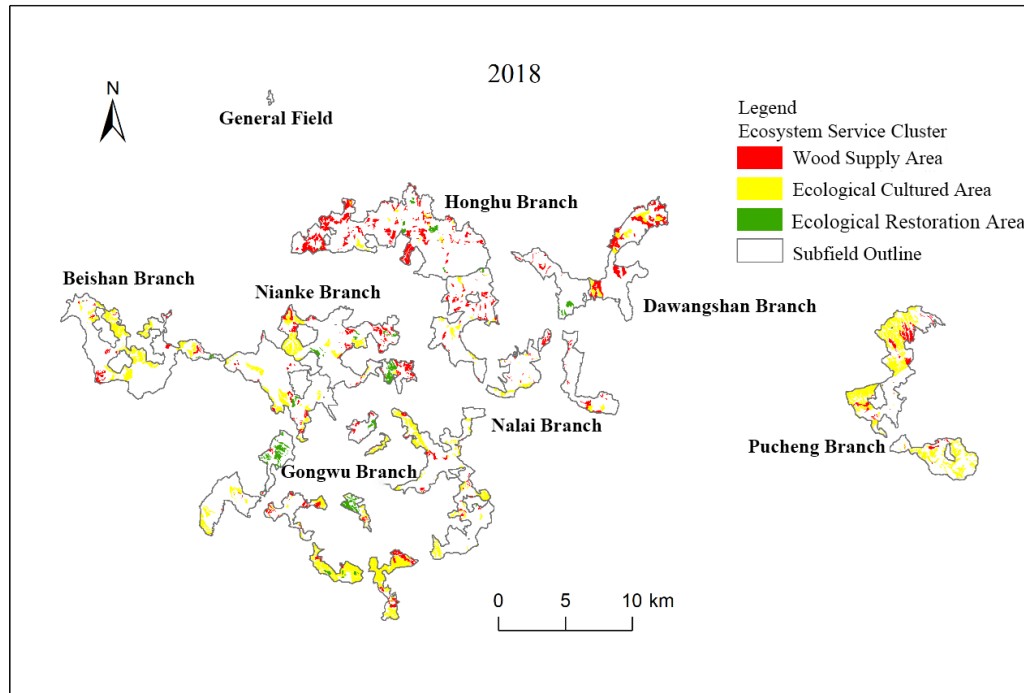

**Figure 6.** Spatial distribution of *PM* plantations' ES bundles in the northern tropical case study area.

**Table 5.** Difference between the ES clusters in the northern tropical case study area of *PM* plantation forests and the mean value of each service in the whole study area.

| Classification | Wood Supply | Carbon Sequestration and Oxygen Release | Water Conservation | Soil Conservation | Number of Small Classes | Service Cluster |
|---|---|---|---|---|---|---|
| C1 | −0.0530 | 0.2397 | −0.5112 | −0.6657 | 797 | Wood supply zone |
| C2 | 0.3978 | 0.4113 | 0.5337 | 0.7719 | 733 | Ecological nourishment zone |
| C3 | −1.3903 | −2.8203 | 0.2009 | −0.0550 | 178 | Ecological restoration zone |

## 5. Discussion

### 5.1. Wood Supply

There were differences in the spatial distribution of the timber provision value of the *PM* plantations in the northern tropical case study area, and the variability was mainly influenced by the year of planting and the hydrothermal conditions, combining the results of this study with those of other scholars [51]. The timber provision capacity of the *PM* plantation forests of different ages was in accordance with the rule that the older the year the greater the timber provision capacity [52]. Therefore, the cultivation cycle was extended to increase the stock of the *PM* plantation forests, and on the basis of this scientific result different silvicultural techniques, such as reasonable density control and mixed planting category control, were carried out for the different forest age stages to increase the timber provision capacity of *PM* [15,53–55].

### 5.2. Carbon Sequestration and Oxygen Release

With the increase in the forest age, the carbon sequestration and oxygen release capacity of *PM* plantations in the northern tropical case study area was enhanced, which was mainly because the carbon sequestration and oxygen release capacity depended on the accumulation of stand productivity, and the carbon sequestration and oxygen release value of *PM* plantations was enhanced with an increase in stand volume, whereas productivity accumulation was enhanced with an increase in forest age, which were similar to the findings of H. Tian et al. [56].

### 5.3. Soil Conservation

The large soil conservation value per hectare in *PM* plantations in the northern tropical case study area was mainly related to the rainfall erosion force and soil cover degree. The soil types in the study area were mainly red soil and lateritic red soil, both of which are clayey soils with small intergranular spaces and interlocking voids that form relatively tortuous capillary channels, which have good water retention due to fact of slow water infiltration but poor water permeability, and do not easily form agglomerate structures, which can easily cause soil erosion. The soil conservation value of the *PM* plantation forests increased with an increase in the forest age, similar to the findings of other scholars [57,58], which plays a crucial role in soil erosion in southern hilly areas due to the large topographic relief.

### 5.4. Water Conservation

During the succession from young to mature forests, the water conservation capacity increased and the water conservation value increased, which are similar to the results of H. Li et al. [59], who studied the gradual increase in the integrated water content capacity of the ecosystem in the evolution of *PM* plantation forests from young to mature forests. The water conservation value of the over-mature forests was the smallest among the different ages of the *PM* plantations, which was because the magnitude of the water conservation was mainly influenced by both rainfall and evapotranspiration [44,60], and with an increase in vegetation cover the evapotranspiration increased while the water content capacity decreased, which are similar to the results of other scholars and studies [57,58]. The experimental observation of water source connotation is more complex and limited by conditions; this study looked for data on the relevant parameters from the literature and

compared them with other studies on rainfall in areas that are geographically close to the application conditions [57]. In future studies, further stand-scale localization observations can be carried out to make the model parameters more accurate.

*5.5. Ecosystem Functional Zoning of the PM Plantation Forests*

Faced with the key services provided by the *PM* plantation forests, such as wood supply, carbon sequestration and oxygen release, water conservation and soil conservation, it was proposed that the above services can be reflected by the service zoning of *PM* plantation forests, and the management of forest ecosystems has changed from relying on traditional empirical subjective decisions in the past to informative, digital and intelligent diverse ecosystem management decisions [61]. The combination of its set of methods for simulating the values of the ESs and the functional zoning methods has an important impact on regional resource management and is of great significance in participating in forest management in addition to guiding the processes of ecological conservation and resource development as well as utilization.

In this study, based on the characteristics and spatial benefits of the respective ESs in small groups, the distribution areas of the *PM* plantation forests in the northern tropical case study area were divided into wood supply, ecological nourishment, and ecological restoration zones. A spectral clustering approach was used to plan the zoning to manage the *PM* plantation forests with four services in a bottom-up approach, tapping into the different management strategy needs of the *PM* plantation forests rather than treating them as homogeneous management zones, which is in line with the general direction of forest ecosystem management [61]. Service zoning according to the values of ESs is important basic work for the coordinated ecological and economic development of *PM* plantation forests in terms of scientific decision making (from experience to science, from perception to abstraction), quantitative management (from qualitative to quantitative), the rationalization of resource development, and the informatization of the management process. It can effectively support forestry units to manage *PM* plantation forests, understand the multiple service characteristics of *PM* plantation forests, and guide a path towards the co-construction of ecological security as well as economic benefits.

## 6. Multi-Objective Management Strategy

In this study, based on the different service characteristics within the service cluster, the northern tropical *PM* plantation forest area was divided into a wood supply zone, ecological nourishment zone, and ecological restoration zone, which can focus more on wood supply and ecological nourishment. The following suggestions are made for the multi-objective management of *PM* plantation forest resources in the camping unit based on their characteristics.

*6.1. Wood Supply Zone*

The dominant basic service in the wood supply zone is timber provision, and other services should be reasonably maintained on the basis of ensuring that the demand for timber is met as the management objective; the utilization and conservation strategy of the *PM* plantation forests in this area focuses on the following:

- Strengthen the high-quality construction of wood supply areas of the *PM* plantations [62] and improve the timber supply capacity of *PM* plantations. The construction of fast-growing and productive *PM* forests can effectively select high-quality container seedlings, improve the limiting factors of the forest land, and improve the quality of the timber volume of the *PM* plantation forests, which are important for guaranteeing the demand for the timber volume of *PM*. Strengthen the basic construction of the wood supply area of the forestry units, emphasize the productive and driving nature of fast-growing, productive *PM* forests, and explore development ideas for the wood supply area by combining them with regional characteristics.

- Plantation forest orientation breeding and improvement: Faced with the different timber needs of the *PM* plantation forests, established *PM* seed source test forests and offspring determination forests were fully utilized to screen out management and management techniques for the planting and replanting of *PM* plantation forests according to the excellent seed sources and excellent family lines in different directions, such as by timber, fat, and pulp [63,64]. For different *PM* plantation cultivation objectives, reasonable breeding techniques, density control techniques, seedling cultivation techniques, nurturing techniques, and fertilization techniques were selected.
- For the reasonable planning of the harvesting scale and post-harvesting treatment measures, consider the timber output as a long-term forest cultivation project; control a reasonable harvesting scale and effectively protect the water conservation and soil conservation services of the forest land from substantial destruction on the basis of enhancing total timber production. Strictly promote the management method of forest harvesting and renewal; the residues existing in the process of the logging area clearing can be cleared out of the usable timber to make comprehensive use of their value or to use them as fuel wood. If they cannot be processed in time due to the presence of artificial or natural reasons, the decay method can be used. For selective or inter-logging woodlands, the residues will be piled up in the blocks or scattered branches method, and, for all-logging woodlands, the strip piling method can be used. These measures can effectively maintain soil nutrients and play a role in water interception as well as soil preservation, which are conducive to the effective restoration of the ES capacity of the woodland after harvesting.

*6.2. Ecological Nourishment Zone*

The dominant ESs in the ecological containment area are soil conservation and water holding capacity, and the key to the utilization and conservation strategy of the plantation forests of the *PM* in this area are as follows:

- Delineate the basic *PM* plantation forest ecological nourishment zone, accounted for with a red line. The *PM* plantation forests have an irreplaceable role compared with other forest stands, and a reasonable guarantee of the ecological content area ratio of the *PM* plantation forests has an important role in the sustainable development of forestry ESs. Therefore, the basic ratio of the ecological nourishment zone should be delineated to ensure the balance of the quantity and quality of high-quality *PM*, guarantee the high-quality level of soil conservation, water conservation, biodiversity, and carbon sequestration and oxygen release in the *PM* plantation forests, and improve the ecological security of the *PM* plantation forests.
- The adjustment of the vegetation planting structure and the promotion of the conservation planting methods accelerate the near-naturalistic recovery of *PM* plantation forests [65]. *PM* plantation forest soil conservation and other services depend on woodland species, vegetation cover, and soil properties in order to determine the *PM* pure forest mixed planting of mullein, maple, milo row, red vertebrae, Runnan, etc., transforming the *PM* plantation forests with a single-species structure into a mixed forest with a rich variety of species and a reasonable structure, using upper, middle, and lower compound planting patterns, increasing the rainwater retention capacity of the stand, increasing the depth and space of the soil root system's complexity, improving the soil consolidation capacity, and promoting soil nutrient accumulation [66,67]. In the process of the planting, replanting, interplanting, or nurturing of *PM* plantations, attention should be paid to soil maintenance to give full play to the soil nutrients and keep the soil adequately permeable.
- Strengthen comprehensive land remediation strategies and implement land ecosystem environmental remediation projects. Forest land remediation can effectively improve the soil quality of forest land, improve the ecological landscape function of plantation forests, and enhance the soil conservation services of *PM* plantation forests. Compre-

hensive land remediation should be strengthened to improve the vegetation cover of *PM* plantation forests and enhance soil conservation capacity.

*6.3. Ecological Restoration Zone*

The water content capacity was relatively high among the four ESs in the ecological restoration area, which is one of the key services of a forest ecosystem. Compared with the intuitive timber volume provisioning capacity, the water content capacity showed the direct radiation of the forest to the surrounding area. The keys to the utilization and conservation strategies of *PM* plantations in the region are as follows:

- Reasonable thinning and understory replanting for the near-natural transformation of *PM* plantation forests. The scientific measures of thinning and replanting were used to transform *PM* plantations' pure forests into near-natural forests with mixed-age coniferous and broad species [68–70] to make full use of the vertical structure of the stand space, enhance the degree of root space utilization in the soil, improve the stand structure, and enhance stand productivity to improve the soil nutrient status and promote the enhancement of carbon sequestration and oxygen release as well as soil conservation capacity.

- After the transformation, the mountain was closed for reforestation to avoid human interference. The closure of mountains is a way to restore forest vegetation by using the regeneration ability of forests as well as by implementing regular closures of mountains in mountainous areas with suitable natural conditions to prohibit human-made destructive activities, such as land reclamation, grazing, and firewood cutting. Due to the diversity of the targets and methods of closure in China, it is also necessary to carry out detailed research on the natural environmental conditions, socioeconomic conditions, and vegetation structure of an area scientifically before closure work is carried out, as well as to carry out the closure work according to local conditions. This can promote the improvement of forest stand structure, biodiversity, carbon sequestration and oxygen release, and water conservation as well as soil conservation capacities [71,72].

## 7. Conclusions

The four ESs of wood supply, carbon sequestration and oxygen release, water conservation, and soil conservation in the northern tropical case study area showed different spatial and temporal distributions depending on the year of planting, topographic factors, and hydrothermal conditions. The value of each ES per hectare in the northern tropical pine plantation, ranked from the largest to the smallest, was water conservation; carbon sequestration and oxygen release; wood supply; and soil conservation, with the values of the wood supply as well as carbon sequestration and oxygen release per hectare increasing with the age of the plantation.

The total values of ESs in the case study area decreased from 2009 to 2013, with a 55.46% growth rate in the total values of ESs. In terms of the structural composition of the total values of ESs in *PM* plantation forests, the total value of wood supply in the northern tropical case study area decreased by 12.39%, the total value of carbon sequestration and oxygen release decreased by 13.28%, the total value of water conservation increased the most, by 206%, and the total value of soil conservation decreased by 10.77%. In terms of the change in the average service value per hectare of plantation forest, the growth rate of the service value per hectare of plantation forest decreases with an increase in latitude; the value of the northern tropical case study area increases by 99.44%, which is nearly double compared with 2009.

From 2013 to 2018, the total area of *PM* plantation forests in the northern tropical case study area decreased by 3.8%, the total values of ESs decreased by 22.70%, and the average total value per hectare decreased by 19.67%. From 2009 to 2018, the total area of *PM* plantation forests in the northern tropical case study area decreased by 24.98% and

the total ecosystem value increased by 20.18%, with an average total value per hectare of 60.19%.

Based on the assessment of the values of the four ESs, we proposed an ecosystem management strategy for *PM* plantation forests based on the results of previous studies and the group's related research. Firstly, through the publicizing and popularization of science, managers and the public should realize that the ESs of *PM* plantation forests are not only for providing economic products such as timber volume and turpentine, but also services such as carbon sequestration and oxygen release, water conservation, and soil conservation, which constitute the basis of our happy lives, such that people can fully understand the diversity of the ESs of *PM* plantation forests, establish correct multi-service values of *PM* plantation forests, and deepen their awareness of conservation as well as utilization. Secondly, four service types, namely wood supply, carbon sequestration and oxygen release, water conservation, and soil conservation, were selected to build a suitable ES evaluation system for *PM* plantation forests, and the relationship between spatial and temporal scale effects on *PM* plantation forests' ESs was fully considered in order to establish an evaluation system for *PM* plantation forests' ESs. Furthermore, to optimize the layout of *PM* plantation ESs' zoning, we carried out differential planning according to the dominant characteristics of different zoning services and carried out localized *PM* plantation forest ecosystem management strategies via forestry units according to the correlation of their location characteristics as well as socioeconomic conditions. Finally, in light of the changing trend in forestry to pursue both long-cycle timber volume production as well as quality and the near-natural multiservice management of plantation ESs, scientific, standardized, and quantified management measures can be adopted to improve the quality and efficiency of *PM* plantation forests, such as genetic improvements, land preparation effects, fertilization effects, density effects, reasonable interval harvesting, biodiversity enhancement and accelerated near-natural recovery succession, plantation maintenance management, etc.

**Author Contributions:** R.M. and Y.W. conceptualized the framework, acquired the funding and supervised the overall project; Y.W. collected the data; R.M. and Y.W. analyzed the data and wrote the manuscript; Y.W., S.D., J.M., and Y.M. provided modification comments; and Y.W., S.D., J.M. and Y.M. reviewed the final manuscript. All authors have read and agreed to the published version of the manuscript.

**Funding:** This work was supported by the National Natural Science Foundation of China (32260387), the Guangxi Key Research and Development Program (GuikeAB21220057), and the Guangxi Innovation-Driven Development Project (GuikeAA20161002-1 and GuikeAA17204087-7).

**Institutional Review Board Statement:** Not applicable.

**Informed Consent Statement:** Not applicable.

**Data Availability Statement:** The data are contained within the article.

**Acknowledgments:** We thank the staff of the Paiyangshan Forest Farm for their help in field sampling and the graduate students at the College of Life Sciences of Guangxi Normal University for their assistance in the indoor experiments.

**Conflicts of Interest:** The authors declare no conflict of interest.

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
