# Peer review of "Ecosystem Service Evaluation and Multi-Objective Management of Pinus massoniana Lamb. Plantations in Guangxi, China"

_forests, doi:10.3390/f14020213_

Round 1
Reviewer 1 Report
This manuscript assesses the quantification of diverse Ecosystem Services in a Chinese area, supplying various strategies for improving the current forest management. My overall recommendation is that the manuscript is further from deserving publication in its current form. The paper's aims and objectives could fit into this journal's editorial policy. The comments are listed as follows:
• General Comment: The authors must improve the manuscript, beginning with fundamental issues. First, the English language is relatively poor. Second, the wording of some parts is very confusing, and some crucial references are incomplete and come from grey literature. Besides, many methodological issues are not correctly explained, and some hypotheses are not adequately justified. In short, a reader cannot reply the study.
•Abstract: the statement about Chinese plantations is indefinite and does not contribute anything to the study
• Introduction: l 42: circular reference. It is hard to understand.
•l. 45-46: What is the difference between ecological and economic value?
•The authors should better explain the objectives
•I'm afraid I have to disagree with the consideration of oxygen release as a new forest management objective. Only in a few cases can this issue be considered, as the authors can check the literature.
•Materials: it is inconceivable that the authors are not provided the plantations' area.
•What is "forest type II survey". A reader does know their significance.
•Methods: (l. 98). The reference is incomplete, and a reader cannot reply the methodology in another case study.
•l. 102: what is a class II survey?
•l. 105-106. Why is the carbon price equal to a Swedish tax? Why do the authors not follow the Chinese national carbon market or another voluntary market?
•l. 121-122. I disagree with the methodology explained to value timber resources.
•l. 158-159. The same is true for the economic valuation of soil erosion.
•l. 205. What is the VEST methodology?
•l. 388-389. This statement is about methodology and should be moved to a previous Section
•lI miss a Discussion Section.
Minor issues:
•Table 2: All Tables must be self-explained.
Author Response
Please allow me to spend more time to read the literature on carbon sequestration and oxygen release as the goal of forest management.

Reviewer 2 Report
The presented article deals with an interesting topic.For ecosystem services evaluation authors used known methods.
The methodical procedure is correct.
Where I see a weakness:
novelty of manuscript
introduction is very simply
incorrect English wording
i would also think about changing the title of the manuscript (simply and more interesting)
Good luck
Author Response
Thank you very much for your suggestion, I will seriously consider the revision about the title of the paper.

Reviewer 3 Report
Thank you very much for the opportunity to read this text. The topic is very important and research towards a service ecosystem is still needed. However, I must point out at the beginning that the text is very difficult and could perhaps be less technical in view of future citations. The methodological part is particularly difficult. The authors present a lot of data, perhaps it would have been better to divide the material and publish the results in two parts.
I have a few comments
1. keywords should not be a duplication of what is contained in the title.
2. the authors do not explain why they are analysing the years 2009, 2013 and 2018. why were these specific years chosen? There needs to be a further development of the topic in section 2.2 "Source of data".
3. what is meant by corrected remote sensing data (line 103). What did this process consist of?
4. in my opinion, since the specifics of the site exclude the development of cultural services (aesthetics, recreation, forest therapy, etc.) it should have been made clear in the title of the paper that it is about the assessment of selected ecosystem services
5 Please note references. Once the names of the researchers are cited, other times only the letter abbreviations of the names. The form of writing should be standardised.
Author Response

(The authors gave the same response as above.)

Reviewer 4 Report
The article is devoted to an important and actual issue of ecosystem services (ES) assessment and management of forest plantations. Based on the ES assessment, the authors divided the study area into three different zones with specific management recommendations, which can be a very useful example of the management interpretation of ES. Unfortunately, the article has serious flaws that must be corrected before publication.
1. English needs moderate improvement.
2. All the drawings are very small, I can't see anything on them.
3. In the Methods section, it is necessary to briefly describe the method of spectral clustering used in section 5.1
4. In the Results section, in particular in tables 3 to 6, it is highly desirable to indicate the obtained values of ES in physical indicators, and not just only in RMB. This is necessary so that the reviewers, and then the readers, can compare the obtained estimates with the results of other studies.
5. Section 5. It is necessary to explain the dynamics of the identified ES values, in particular:
- Why does wood value per ha decreases for young forests in 2018 (Table 3), but the ES of soil conservation does not decreases (Table 5)?
- Why do the values of the first three ES for medium-aged and mature forests monotonously increasing, while ES of water conservation decreases in 2018?
If the dynamics of indicators is important for the conclusions (which is not clear from the text, see the next comment), it is useful to show them on a single graph for comparison.
6. Section 5. What is the significance of the identified dynamics of ES values for zoning and management recommendations? Otherwise, why is it presented in the article?
7. Section 5.1. It is completely unclear how exactly this zoning was done. It is necessary to describe the method in the Methods section and explain the results here. What bundles of ES do you mean (line 404)?
8. Section 5. The functional area of ecological conservation is named differently four times: ecological conservation (line 401), cultured (fig. 6 and line 437), connotation (line 409), containtment (line 438). One term must be used.
Minor comments
Line 23 - It must be indicated that the monetary value of ES was compared.
Line 59 - What previous studies are mentioned? Are these studies by the authors or other researchers? Need links.
Line 98 - What is this document or study? It's not clear from reference 10
Lines 105-107 - Repeat lines 139-141. From here it is better to remove.
Line 106 - Not everyone knows that the RBM is the yuan. Better explain at first mention.
Line 108 - I did not find in the following text the price used to monetize the soil conservation ES.
Lines 233 and 245 - Why Sargassum pine here?
Lines 470-476 - What "closure" do you mean? Is this some form of banning people from going there? This needs to be explained.
Author Response
Thank you for your comments, I will take them into consideration.

Round 2
Reviewer 4 Report
The authors corrected all these shortcomings. The manuscript may be published
Author Response
The English grammar of the manuscript has been revised.